# Direct and indirect effects of the COVID-19 pandemic on mortality in Switzerland

Julien Riou [1,2], Anthony Hauser [1,2], Anna Fesser[2], Christian L. Althaus [1], Matthias Egger [1,3,4] & Garyfallos Konstantinoudis [5] ✉

The direct and indirect impact of the COVID-19 pandemic on population-level mortality is of concern to public health but challenging to quantify. Using data for 2011–2019, we applied Bayesian models to predict the expected number of deaths in Switzerland and compared them with laboratory-confirmed COVID-19 deaths from February 2020 to April 2022 (study period). We estimated that COVID-19-related mortality was underestimated by a factor of 0.72 (95% credible interval [CrI]: 0.46–0.78). After accounting for COVID-19 deaths, the observed mortality was −4% (95% CrI: −8 to 0) lower than expected. The deficit in mortality was concentrated in age groups 40–59 (−12%, 95%CrI: −19 to −5) and 60–69 (−8%, 95%CrI: −15 to −2). Although COVID-19 control measures may have negative effects, after subtracting COVID-19 deaths, there were fewer deaths in Switzerland during the pandemic than expected, suggesting that any negative effects of control measures were offset by the positive effects. These results have important implications for the ongoing debate about the appropriateness of COVID-19 control measures.

The COVID-19 pandemic has affected mortality globally through direct and indirect effects. Infection with SARS-CoV-2 directly causes death in a small proportion of infected people, with variations in the infection-fatality ratio (IFR) depending on age[1], socio-economic position[2], vaccination status[3], intensive care unit capacity[4] and several other characteristics of individuals or communities. Deaths directly attributable to SARS-CoV-2 infection may be both laboratory-confirmed deaths and deaths attributable to SARS-CoV-2 without testing (e.g., deaths following a heart attack or stroke[5,6]). Combined with the high transmissibility of SARS-CoV-2, this has resulted in more than 6.5 million laboratory-confirmed deaths globally (as of October 10, 2022)[7], which is likely an underestimate of the true number of deaths directly attributable to SARS-CoV-2 infection. The pandemic has also caused major disruptions in many aspects of social and economic life and may thus have indirectly increased or reduced mortality. Non-pharmaceutical interventions (NPIs) may lead to delays or avoidance of medical care[8,9], increases in substance use and suicidal ideation[10–12], or increases in interpersonal violence[13]. Conversely, stay-at-home orders have led to reductions in mobility and traffic[14] and air pollution levels[15]. Border closures and reductions in social contacts and activities have restricted the circulation of other infectious diseases[16]. The respective importance of the pandemic's positive and negative indirect effects on mortality and the net impact remains unknown.

The overall impact of the COVID-19 pandemic on mortality at the population level, both directly and indirectly, is of great concern to public health but is difficult to quantify. Laboratory-confirmed deaths (i.e., deceased people with a recent positive SARS-CoV-2 test) may underestimate mortality as some deaths will remain unascertained, for example, due to testing policies, shortages, underreporting or overwhelmed health systems[17]. Laboratory-confirmed deaths ignore indirect effects on mortality. The main alternative metric relies on excess mortality estimated from all-cause mortality data, using counterfactual reasoning[18]. The observed number of deaths is compared to what would have been expected had the pandemic not occurred, based on

[1]Institute of Social and Preventive Medicine, University of Bern, Bern, Switzerland. [2]Federal Office of Public Health, Bern, Switzerland. [3]Population Health Sciences, Bristol Medical School, University of Bristol, Bristol, UK. [4]Centre for Infectious Disease Epidemiology and Research, University of Cape Town, Cape Town, South Africa. [5]MRC Centre for Environment and Health, Department of Epidemiology and Biostatistics, School of Public Health, Imperial College London, London, UK. ✉e-mail: g.konstantinoudis@imperial.ac.uk

mortality data from previous years and considering demographic changes and covariates associated with mortality patterns. The approach has the advantage of covering both the pandemic's direct and indirect effects, although phenomena like mortality displacement can limit the interpretability of results[19–21]. Also, estimations of excess mortality depend on model assumptions and methodological choices, such as age-specific population trends[22]. Detailed analyses of causes of death as listed in death certificates can also be used (generally with considerable delay) but suffer from significant limitations, especially regarding ascertaining infectious diseases[23].

There have been many attempts to estimate excess mortality associated with the COVID-19 pandemic in various settings[24–34]. Comparisons of excess mortality with laboratory-confirmed deaths have confirmed that the overall impact of the pandemic on mortality is generally much greater than what is indicated by laboratory-confirmed deaths alone[27,32,34]. Still, a common limitation of these studies is the inability to distinguish between the direct and indirect effects of the pandemic on mortality. This study attempts to overcome this limitation by jointly studying laboratory-confirmed COVID-19-related deaths and excess mortality. We computed the expected number of all-cause deaths by week, age group, and location in Switzerland between February 2020 and April 2022, accounting for the effect of temperature, national holidays, and population changes using a validated statistical approach[35]. We then developed a method to partition all-cause mortality into deaths directly attributable to SARS-CoV-2 infection and deaths indirectly attributable to the pandemic as a result of all the changes in health, health care, and living or working conditions associated with the SARS-CoV-2 pandemic in Switzerland. We could thus examine the completeness of ascertainment of COVID-19-related deaths and the indirect effects of the pandemic on all-cause mortality in Switzerland.

## Results

We observed a total of 156,193 deaths from all causes in Switzerland from February 24, 2020, to April 3, 2022, compared to an expected 142,408 (95% credible interval [CrI]: 138,044 to 149,125) had the pandemic not occurred. This translates into 13,786 (95% CrI: 7068 to 18,149) excess all-cause deaths over the pandemic period, a relative increase of 9.7% (95% CrI: 4.7 to 13.1). There were three periods of substantial relative excess mortality: 7.3% (95% CrI: 3.8 to 10.8) during phase 1, 33.9% (95% CrI: 26.4 to 41.4) during phase 3 and 15.9% (95% CrI: 8.3 to 22.8) during phase 6 (Table 1). There were slightly fewer all-cause deaths than expected during phase 4, with a relative excess mortality of −4.3% (95% CrI: −9.9 to 0.2). The age groups affected most by excess mortality were those over 70 years of age (Fig. 1A, B).

A total of 13,130 laboratory-confirmed COVID-19-related deaths were reported during the study period. Weekly counts of laboratory-confirmed deaths generally aligned with estimates of excess all-cause mortality in Switzerland (Fig. 2), with a correlation coefficient of 0.89 (95% CrI: 0.85 to 0.92). The estimate of $\beta_1$ was 1.38 (95% CrI: 1.20 to 1.54), suggesting that there were, on average, 38% (95% CrI: 20 to 54) more deaths directly attributable to COVID-19 than laboratory-confirmed deaths during the period or that ascertainment proportion was 72% (95% CrI: 65 to 83) (Table 1). Given the 13,130 laboratory-confirmed deaths over the period, this implies that the total number of deaths directly attributable to COVID-19 in Switzerland until April 3, 2022, can be estimated at 18,177 (95% CrI: 15,820 to 20,283) deaths.

After accounting for deaths directly attributable to COVID-19, the observed number of all-cause deaths was slightly lower than expected based on historical trends. This deficit is quantified by $\beta_2$, estimated at 0.96 (95% CrI: 0.92 to 1.00), indicating −4% (95% CrI: −8 to 0) fewer all-cause deaths than expected during the COVID-19 pandemic after adjusting for the direct effect of SARS-CoV-2 infections on mortality. Still, the data are compatible with no indirect beneficial effect.

The coefficients $\beta_1$ and $\beta_2$ varied across age groups and time periods. The alignment between excess mortality and laboratory-confirmed deaths was particularly noticeable in age groups 70–79 and 80 and older (Fig. 3A), and during phases 1, 3, and 6 (Supplementary Fig. S1). Variation in the relative number of deaths directly attributable to COVID-19 for each laboratory-confirmed death ($\beta_1$) by age group suggests that more deaths were not ascertained in age groups 70–79 and 80+, while the data were compatible with 100% ascertainment ($\beta_1 = 1$) in age groups below 70, where fewer deaths were reported (Fig. 3B). $\beta_1$ was estimated around 1.5 during phases 1 and 3 and around 2 during phase 6, suggesting an ascertainment proportion of COVID-19 deaths during large epidemic waves ranging between 50 and 66% (Fig. 3B). This estimate is less precise during periods of low epidemic activity (phases 2, 4, 5, and 7), and remains compatible with 1 (perfect ascertainment). The relative deficit in all-cause deaths after accounting for deaths directly attributable to COVID-19 ($\beta_2$) was more pronounced in age groups 40–59 and 60–69 with estimates of −12% (95% CrI: −19 to −5) and −8% (95% CrI: −15 to −2), respectively. The deficit was also concentrated in phases 1, 3, and 4 (Fig. 3C). Estimates of $\beta_1$ and $\beta_2$ across administrative regions show generally homogeneous results for the whole of Switzerland (Supplementary Fig. S2 and Table S2).

We conducted a sensitivity analysis where the population over time was corrected by excess mortality as it occurred. This resulted in higher estimates of excess mortality, with about 1000 additional excess deaths but had little effect on the estimates of $\beta_1$ and $\beta_2$ (Supplementary text S1.3).

## Discussion

We examined the patterns of all-cause mortality in Switzerland from the diagnosis of the first COVID-19 case at the end of February 2020 until April 2022. We compared the excess mortality with laboratory-

**Table 1 | Mean and 95% credible intervals for expected and excess number of all-cause deaths, relative excess all-cause mortality, and number of observed all-cause and laboratory-confirmed COVID-19-related deaths by seven epidemic phases between February 2020 to April 2022**

| Phase[a] | Expected | Observed | Excess | Relative excess | Laboratory |
|---|---|---|---|---|---|
| 1 | 19,376 (18,767 to 20,033) | 20,791 | 1415 (758–2024) | 7% (4–11) | 1725 |
| 2 | 19,180 (18,440 to 20,042) | 19,103 | −76 (−939 to 663) | −0% (−5 to 4) | 104 |
| 3 | 27,004 (25,569 to 28,604) | 36,157 | 9154 (7553 to 10,588) | 34% (26–41) | 7652 |
| 4 | 23,386 (22,320 to 24,834) | 22,369 | −1017 (−2465 to 49) | −4% (−10 to 0) | 895 |
| 5 | 19,175 (18,284 to 20,223) | 20,007 | 832 (−216 to 1723) | 4% (−1 to 9) | 380 |
| 6 | 13,036 (12,298 to 13,944) | 15,105 | 2070 (1161–2807) | 16% (8–23) | 956 |
| 7 | 21,370 (20,067 to 22,894) | 22,661 | 1291 (−233 to 2594) | 6% (−1 to 13) | 1418 |

[a]Phase 1: February 24, 2020 to June 7, 2020, phase 2: June 8, 2020 to September 27, 2020, phase 3: September 28, 2020 to February 14, 2021, phase 4: February 15, 2021 to June 20, 2021, phase 5: June 21, 2021 to October 10, 2021, phase 6: October 11, 2021 to December 19, 2021 and phase 7: December 20, 2021 to April 3, 2022.

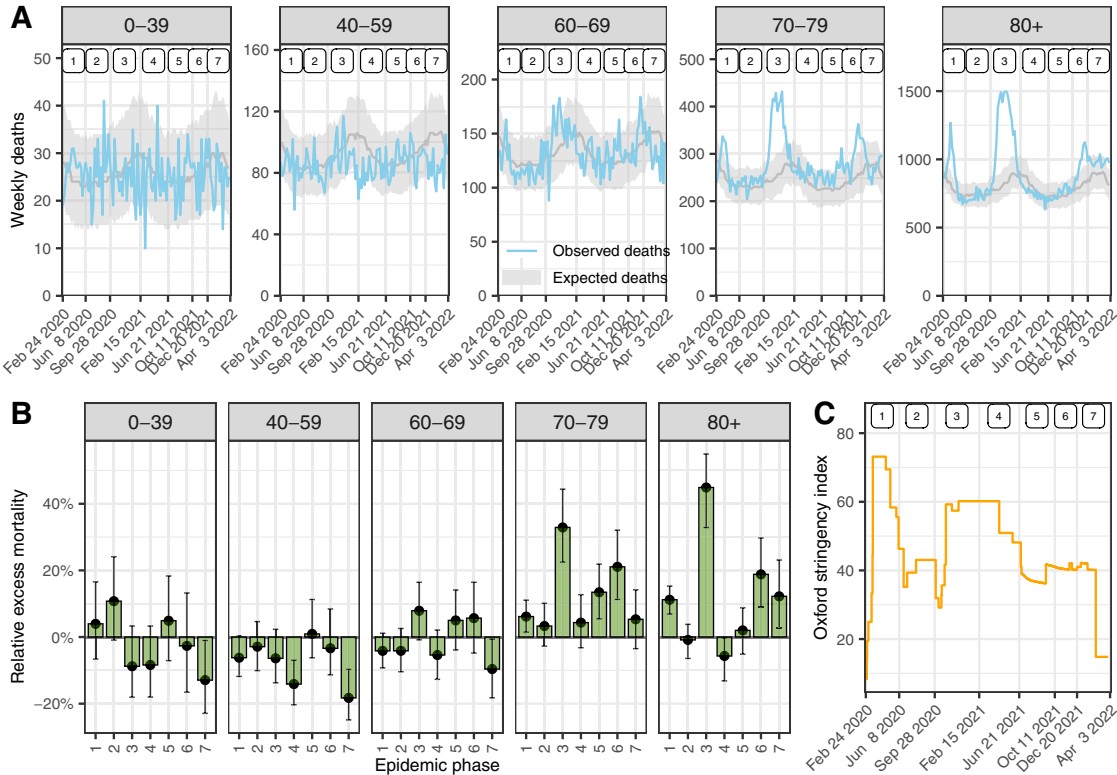

**Fig. 1 | Observed and expected number of deaths, relative excess mortality, and the Oxford stringency index. A** Observed and expected number of weekly deaths by age group in Switzerland from February 2020 to April 2022. Model-predicted expected deaths are shown with a median and 95% credibility interval. Numbers at the top indicate epidemic phases 1–7. **B** Estimated relative excess mortality by seven epidemic phases from February 2020 to April 2022 and five age groups. Medians with 95% credible intervals (error bars) are shown. See Online Supplementary Table S3 for the size of each group. **C** Timeline of the Oxford stringency index in Switzerland[50]. $n = 588,739$ during January 2011 and December 2019 and $n = 156,193$ observed all-cause deaths during February 2020 and April 2022 were used to derive the above statistics.

confirmed COVID-19-related deaths and partitioned excess mortality into the excess directly attributable to COVID-19 and excess or under mortality indirectly attributable to the pandemic. We found that the estimated number of deaths directly caused by COVID-19 was about 1.4 times higher than the number of laboratory-confirmed deaths. In other words, only about 70% of COVID-19-related deaths were ascertained. Overall, COVID-19 was directly responsible for an estimated 18,000 deaths during the study period, during which only around 13,000 laboratory-confirmed COVID-19-related deaths were reported. Finally, the pandemic may have had an indirect beneficial effect on mortality in age groups 40–69.

Our study has several strengths. Detailed data on the population structure, mortality, weather, and national holidays from the 10 years before the COVID-19 pandemic allowed us to estimate what mortality would have been in 2020–2022 had the pandemic not occurred. Estimation of excess all-cause mortality during the pandemic period by time, space and age thus became possible. Temperature, holidays, and population are certainly the most important determinants of changes in all-cause mortality[36–39], but the modelling framework we developed also accounts for unknown factors that may vary in space and in time (both in the seasonal and long-term)[29,40], resulting in a model with high predictive ability (Supplementary Text S1.1–1.2). It correctly handles uncertainty from the different data sources and propagates it to the final estimates. The approach may be used in any setting with reliable reports of all-cause mortality and laboratory-confirmed deaths. Further, we developed a novel statistical method to differentiate between deaths directly attributable to SARS-CoV-2 infections and deaths caused or prevented indirectly by the pandemic.

Our study also has several limitations. As with other studies of excess mortality[24–34], we assume that the only difference between the

2015–2019 and 2020–2022 time periods other than these covariates is the presence or absence of the COVID-19 pandemic. Therefore, any difference in mortality between these two periods beyond random variation is attributed to the pandemic, either directly or as a consequence of changes in health, health care, and living or working conditions associated with the SARS-CoV-2 pandemic in Switzerland. We ignored the potential impact of other factors on mortality that may have occurred in 2020–2022 but not in 2011–2019. In particular, the study period excluded the heat wave in the summer of 2022. Our modelling framework cannot easily distinguish between reductions in mortality and mortality displacement between epidemic phases. The standard approach to computing excess mortality ignores the changes in population caused by the excess deaths themselves, resulting in an overestimation of the expected number of deaths towards the end of the study period, and thus an underestimation of excess mortality. A sensitivity analysis correcting for this issue resulted in an additional 1000 excess deaths over the full period, but this did not impact our estimates of $\beta_1$ and $\beta_2$. Information about the cause of death was not available for the full study period but would have helped understand the mechanisms of the indirect beneficial effect of the COVID-19 pandemic on mortality. We also only considered the short-term effects of COVID-19 on mortality, while SARS-CoV-2 infection may lead to increased mortality from cardiovascular, cancer or respiratory system causes of death in the following months[6]. We assumed deaths with a positive SARS-CoV-2 test were caused by COVID-19, although the infection could be incidental in some cases (the median delay from test to death was 10 days, interquartile range of 6–15). Such misclassification would only concern a small proportion of deaths with laboratory-confirmed infection, as was shown in cause-of-death data[41]. Autopsies of patients dying in hospital following a positive SARS-CoV-2 test also

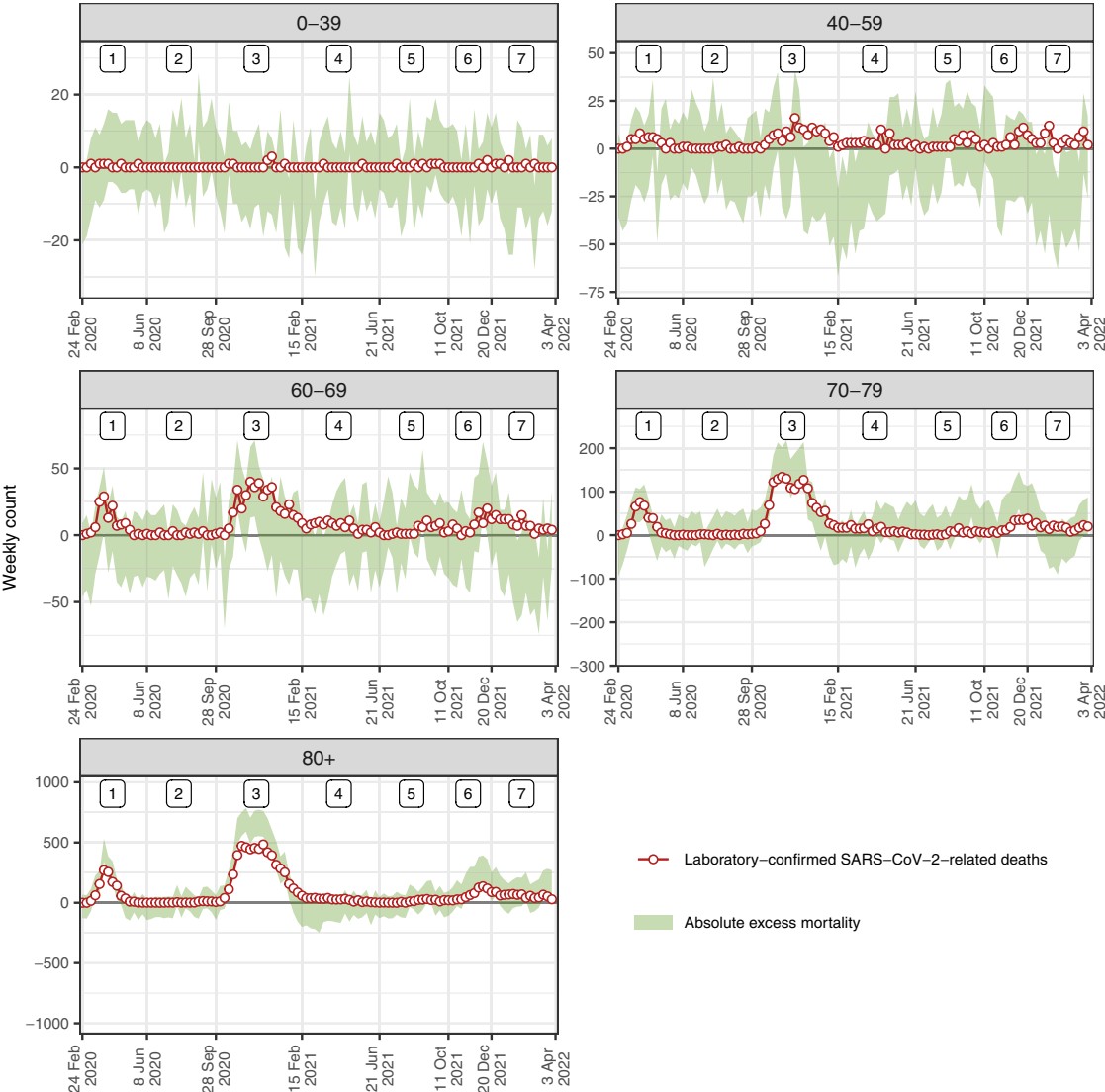

**Fig. 2 | Excess all-cause and laboratory-confirmed COVID-19-related deaths.** Weekly counts of excess all-cause deaths (95% credibility intervals) and of laboratory-confirmed COVID-19-related deaths between February 24, 2020 and April 3, 2022 in Switzerland by five age groups. Numbers at the top indicate epidemic phases 1–7.

suggest that during the study period, causes of death were generally directly related to COVID-19[42]. Still, this potential bias may lead to an underestimation of the ascertainment proportion in our analysis. Finally, we did not stratify by sex in this study. In a previous analysis, we found only small differences in excess deaths between sexes[29].

Our estimates of excess mortality during the COVID-19 pandemic in Switzerland are consistent with other analyses. The Federal Office of Statistics reported excess mortality above 10% for January 2020 to August 2021[33], higher than the 9.7% estimated in the present study for the period up to spring 2022. A multi-country study estimated excess mortality at 13,000 deaths in Switzerland from March 2020 to June 2022[28,43] and another one at 15,500 (14,000 to 17,000) for 2020 and 2021[32]. In our study of five European countries, we estimated an excess mortality of 8% in males and 9% in females for Switzerland during the first year of the pandemic[29]. WHO estimates for Switzerland in 2020 and 2021 were lower (8200 excess deaths), but there were problems with the WHO estimates[30,44]. None of these studies attempted to quantify the direct and indirect effects of the pandemic on mortality. In line with our estimates, a study in California reported a 78% ascertainment proportion of diagnosed COVID-19 deaths but did not quantify the indirect effect of the pandemic[45]. This is the first study

that aims to quantify both the direct and indirect effects of the pandemic on mortality.

We found that COVID-19 caused about 1.4 times more deaths than were laboratory-confirmed, in line with a recent study estimating this ratio at 1.29 (1.16–1.42) for Switzerland[34]. Estimates varied widely between countries. For example, the ratio was 0.57 (0–1.25) for Norway but 150 (140–162) for Nicaragua[34]. Differences could be attributable to local healthcare and surveillance systems, testing capacity, and methodological differences in collecting mortality data and estimating excess mortality. Recently published data on causes of death in 2020 and the first half of 2021 provided a partial external validation of our results[41]. According to these data, the number of deaths with COVID-19 as a cause was 2142 from February to May 2020 (24% more than laboratory-confirmed deaths) and 10,650 from September 2020 to February 2021 (36% more than laboratory-confirmed deaths). We found markedly lower ascertainment during periods of high epidemic activity, suggesting shortcomings in testing, or reporting even in Switzerland, a high-income country. Under-ascertainment was concentrated in older age groups, suggesting incomplete ascertainment in retirement and nursing homes, in line with other reports[17]. During the first epidemic phase, testing capacities were limited, which might have

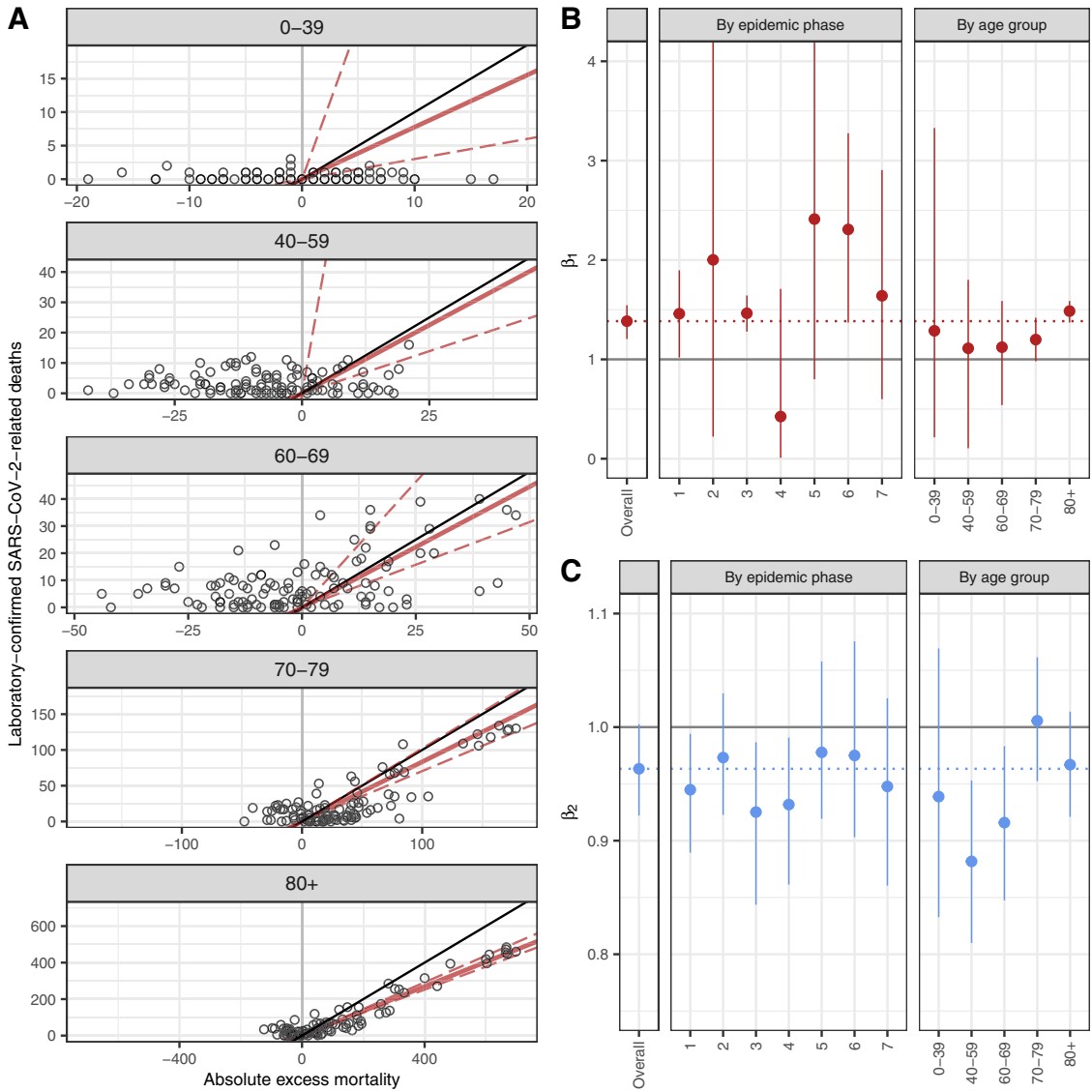

**Fig. 3 | Direct ($\beta_1$) and indirect ($\beta_2$) effects of the COVID-19 pandemic and all-cause mortality by age and epidemic phase. A** Association between weekly laboratory-confirmed COVID-19-related deaths and absolute excess mortality by age group. The black line shows the slope of association corresponding to a 1 to 1 relation. The red lines show the association estimated with the model (corresponding to the $\beta_1$ coefficients shown in panel (**B**), the full line represents the point estimate and the dashed lines the lower and upper bounds of the 95% credible interval). **B** Estimates of $\beta_1$, the additional number of deaths to be observed for each unit increase in laboratory-confirmed deaths, after adjusting for the expected number of all-causes deaths given historical trends. The error bars denote the 95% Credible Intervals. See Online Supplementary Table S3 for the size of each group. **C** Estimates of $\beta_2$, the additional number of deaths to be observed for each unit increase in the expected number of all-cause deaths, after adjusting for the direct effect of SARS-CoV-2 infections. The error bars denote the 95% credible intervals. See Online Supplementary Table S3 for the size of each group. $n$ = 156,193 number of observed all-cause deaths were used to derive the above statistics.

affected the ascertainment of COVID-19-related deaths. However, $\beta_1$ in epidemic phase 1 did not substantially differ from the overall estimate. While testing increased over time in Switzerland, the targeting of tests to vulnerable population groups may have simultaneously decreased, so that the ascertainment of severe cases remained relatively stable in the first four phases. The lower ascertainment towards the end of the study period might be explained by reduced testing once vaccines became available.

We found that in Switzerland, the COVID-19 pandemic probably had an indirect beneficial effect on mortality. A potential explanation is mortality displacement or the "harvesting effect", where COVID-19 precipitated deaths that would have occurred anyway[19–21,46]. The short period of negative excess mortality observed at the beginning of phase 4, just after the largest epidemic wave, could be explained by this harvesting effect. However, this can only be a partial explanation: the overall deficit of deaths was concentrated in the younger

age groups and not in the over 70 years old, where mortality displacement typically occurs. Therefore, the mortality deficit is probably due to the pandemic's indirect effects, such as reductions in mobility, road traffic, air pollution and sports activities. The fact that the deficit was more pronounced during phases 1, 3, and 4 when control measures were most stringent[47], supports this interpretation. The concentration of the mortality deficit in the younger age groups also argues against an important role of a reduced prevalence of other pathogens. For example, influenza leads to mortality in the older age groups, the lack thereof would therefore be expected to result in a mortality deficit in these age groups. Influenza seasons during 2011–2019 were accounted for by the spatiotemporal component of the model. In any case, we find no evidence for an overall detrimental effect of control measures on mortality. However, we cannot rule out harmful effects, such as delay or avoidance of medical care[8,9], increases in drug use and suicidal ideation[10,12], increases in

interpersonal violence[13] and other risks associated with economically precarious living conditions. These effects were likely mitigated by government policies. The short-time work support scheme, which benefited many workers (1.3 million at the peak in April 2020), probably prevented many redundancies. Indeed, the increase in unemployment in 2020 was relatively modest (from 2.3% in 2019 to 3.1%), and the unemployment rate decreased again in the following years[48,49].

Our results are not readily applicable to other countries. Switzerland is a high-income country with a relatively old but healthy population. The stringency of control measures was relatively mild compared to other European countries[50]. While any harmful indirect effects of control measures appear to have been more than compensated in Switzerland, further research is required to quantify indirect effects in other countries. Our approach can be used for this purpose, but it does not elucidate the pathways leading to an increase or a decrease in mortality. Further research using cause-specific mortality data is needed to answer this question. Studies over many years are required to gauge the long-term effects on mortality, if any, the COVID-19 pandemic.

In conclusion, shortcomings in testing coverage caused considerable underestimation of COVID-19-related deaths in Switzerland, particularly in older populations. Although COVID-19 control measures may have negative effects (e.g., delays in medical care or mental health impairments), we note that after subtracting deaths directly caused by SARS-CoV-2 infections, there were fewer deaths in Switzerland during the pandemic than expected. This deficit cannot be attributed to a displacement of mortality, as it was observed mainly in the 40–69 age group. This suggests that any negative effects of control measures on mortality were more than offset by the positive effects. These results have important implications for the ongoing debate about the appropriateness of COVID-19 control measures.

## Methods

### Data sources
We retrieved population data in Switzerland for the pre-pandemic years 2011–2019 from the Federal Statistical Office (FSO). Data were aggregated by age group (in five groups: 0–39, 40–59, 60–69, 70–79 and 80 and older), sex (two groups) and administrative region (26 cantons). Data on all-cause deaths were also obtained from the FSO. These consisted of counts of deaths from any cause by age, sex, and canton for each week using the date of deaths from 2011 to 2019, and afterwards for each week up to April 3, 2022. We used data on ambient temperature from the European Centre for Medium-Range Weather Forecasts Reanalysis version 5 (ERA5) reanalysis data set[51] and on national holidays[52]. Daily mean ambient temperature between 2011 and 2022 at 0.25° × 0.25° resolution was aggregated by taking means per week and canton. Holidays were defined weekly for each canton (1 if there was at least one cantonal holiday, 0 otherwise).

The reporting of laboratory-confirmed COVID-19-related deaths has been mandatory in Switzerland since February 2020. The records are kept at the Federal Office of Public Health (FOPH) and are available online. Data include age, sex, canton of residence, and the date and type of the positive SARS-CoV-2 test. Dates were grouped into seven epidemic phases by the FOPH using the lowest counts of reported cases: February 24, 2020 to June 7, 2020 (phase 1); June 8, 2020 to September 27, 2020 (phase 2); September 28, 2020 to February 14, 2021 (phase 3); February 15, 2021 to June 20, 2021 (phase 4); June 21, 2021 to October 10, 2021 (phase 5); October 11, 2021 to December 19, 2021 (phase 6) and December 20, 2021 to April 3, 2022 (phase 7).

### Population trends model
We used population size on December 31, 2010 to 2019 by age group, sex and canton to predict population sizes in each stratum and week of the entire study period (January 1, 2020 to April 3, 2022) in a two-step procedure. First, we fitted a Poisson regression model to population data from 2011 to 2019. This model included a linear yearly trend, a fixed effect by sex, and independent random effects by week (for seasonality), age group and canton. We compared different models using higher interactions and yearly linear trends that vary by space, age, and sex. Model comparison using a cross-validation scheme excluding the last three years of available data (2017–2019) determined that the best model included all possible two-way interactions between age, canton, and week, and an overdispersion parameter. We obtained posterior distributions of the population in each stratum for December 31, 2020, 2021 and 2022, under the counterfactual scenario that the pandemic did not occur. In the second step, we used linear interpolation to obtain weekly population size (estimates, with uncertainty). Supplementary Text S1.1 provides further details.

### Expected deaths model
We estimated the expected number of all-cause deaths for each week between February 24, 2020, the day of the first confirmed COVID-19 case in Switzerland, and April 3, 2022 by age, sex and canton of residence using the historical data (2011–2019) and expanding a previously proposed model[29]. We used Bayesian spatiotemporal models accounting for population trends and including covariates related to temperature and national holidays. To account for uncertainty in population estimates, we applied the model multiple times over the samples of the posterior distributions of the population predictions. Since the effect of temperature on all-cause mortality is expected to be U-shaped[53], we used a random walk of order 2 to allow for a flexible fit. We accounted for seasonality using a random walk of order 1 at the weekly level, and for exceptional events using week-level independent random effects. We accounted for long-term trends with a linear slope at the yearly level, and for spatial autocorrelation using conditional autoregressive priors. We modelled spatial autocorrelation using an extension of the Besag–York–Mollié model, allowing for a mixing parameter measuring the proportion of the marginal variance explained by the spatial autocorrelation term[54,55]. The model has been internally validated and found to be unbiased and to have high predictive accuracy. The weaker correlation in the younger age groups was expected as the Pearson correlation is not suitable for small counts. Coverage and bias indicate high predictive accuracy also in these age groups. We used the fitted model to obtain posterior distributions of the expected number of all-cause deaths by age group, sex and canton in each week between February 24, 2020 and April 3, 2022. Estimates of excess mortality (with uncertainty) were then obtained by subtracting the expected (across the posterior samples) from the observed all-cause deaths in each stratum. Supplementary Text S1.2 provides further details and the results of the internal cross-validation. One limitation of the proposed approach is that it ignores changes in the population structure caused by the excess deaths themselves, leading to an overestimation of population and expected deaths, and, thus, an underestimation of excess towards the end of the study period. We corrected this bias in a sensitivity analysis (Supplementary text S1.3).

### Separation model
We first studied the alignment between excess mortality and laboratory-confirmed COVID-19-related deaths using Pearson's correlation coefficient (applied across the posterior samples of excess mortality to propagate uncertainty). We then developed a method to partition the number of all-cause deaths observed in the pandemic period based on 1) the number of laboratory-confirmed COVID-19-related deaths and 2) the number of expected deaths given historical trends. We included multiplicative parameters to measure the respective contributions of these two quantities. We used a Poisson regression model with an identity link and no intercept term of the

form:

$$O_t \sim \text{Poisson}(\beta_1 L_t + \beta_2 E_t + u_t), \qquad (1)$$

where $O_t$ is the observed number of all-cause deaths on week $t$, $L_t$ is the number of laboratory-confirmed COVID-19-related deaths, $E_t$ is the expected number of all-cause deaths given historical trends, and $u_t$ is a normally distributed overdispersion term centred at zero.

Within this formulation given in (1), $\beta_1$ is the number of all-cause deaths for each unit increase in laboratory-confirmed deaths, after adjusting for the expected number of all-cause deaths given historical trends. That means that under perfect case ascertainment $\beta_1 = 1$. If $\beta_1 > 1$, then we observe a greater number of deaths attributed to SARS-CoV-2 infections compared with the number of laboratory-confirmed deaths. The ascertainment proportion of COVID-19-related deaths is obtained by $1/\beta_1$. This relies on the assumption that when there is at least one laboratory-confirmed death in a week, then the excess in observed all-cause deaths can be directly attributed to COVID-19. In a similar way, $\beta_2$ is the number of all-cause deaths for each unit increase in the expected number of all-cause deaths after adjusting for the direct effect of COVID-19. We expect $\beta_2 = 1$ when the net effect of the pandemic-related behavioural, societal and health system changes on all-cause deaths is zero. The estimate of $\beta_2$ can thus be interpreted as a measure of the indirect effect of the pandemic on mortality. If $\beta_2 < 1$, then there were fewer all-cause deaths than expected after removing the direct effect of COVID-19, which could imply an indirect protective effect of all changes and control measures associated with the pandemic. Estimates of $\beta_1$ and $\beta_2$ thus provide a way to understand the interplay between laboratory-confirmed COVID-19-related deaths and excess all-cause deaths and allow to differentiate between direct and indirect consequences of the COVID-19 pandemic on mortality. Supplementary Text S1.4 provides further details on model specification and choices of the priors.

We extended the model presented above to examine these associations by phase (from 1 to 7 as defined by the FOPH), by age group (0–39, 40–59, 60–69, 70–79 and 80+ years old), and by area (26 cantons). To this end, we introduced multiple $\beta_1$ and $\beta_2$ for each phase, age group or area separately, with the additional constraint of a multilevel structure allowing a smoothing towards the global mean of the estimator[56]. To propagate the uncertainty of the expected number of deaths, we fitted the above models using 200 samples of the posterior distribution of the expected number of deaths. We then combined the resulting posterior samples of $\beta_1$ and $\beta_2$.

All inferences were done in a Bayesian framework. Posterior distributions were approximated by samples, and summarised by their median, 2.5% and 97.5% percentiles to obtain point estimates and 95% credible intervals (95% CrI). The population and expected deaths models were implemented in R-INLA[57], and the separation model in NIMBLE[58]. We performed the analysis using the R software and the code is available online[59].

### Ethics
The study is about secondary, aggregated, and anonymised data, so no ethical permission is required.

### Reporting summary
Further information on research design is available in the Nature Portfolio Reporting Summary linked to this article.

## Data availability
Data on population and all-cause mortality are freely available on the FSO website at https://www.pxweb.bfs.admin.ch/pxweb/en/ and https://www.bfs.admin.ch/bfs/en/home/statistics/population/births-deaths.html. Data on laboratory-confirmed deaths are freely available online (https://www.covid19.admin.ch/en/overview). Daily mean temperature data are freely available online (https://cds.climate.copernicus.eu/cdsapp#!/dataset/reanalysis-era5-single-levels?tab=overview).

## Code availability
The code is available on GitHub.

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

## Acknowledgements

This study would not have been possible without the extraordinary efforts of the data science team at the Federal Office of Public Health. We also thank Rolf Weitkunat (Federal Statistical Office) for his helpful comments. This study was funded by the SFOPH and the Swiss National Science Foundation (grant 189498). C.L.A. acknowledge funding from the EU's Horizon 2020 research and innovation programme (project EpiPose, 101003688). G.K. is supported by an MRC Skills Development Fellowship [MR/T025352/1].

## Author contributions

J.R., A.H., and G.K. conceived the study. J.R. and G.K. drafted the first version of the paper, did all statistical analyses, and take responsibility for the integrity of the data and the accuracy of the data analysis. M.E., C.L.A., and A.F. revised the paper. All authors contributed to the interpretation of data and read and approved the final paper.

## Competing interests

The authors declare no competing interests.
