## [Peer Review File · Nature Communications]

Direct and indirect effects of the COVID-19 pandemic on mortality in SwitzerlandREVIEWER COMMENTS

Reviewer #1 (Remarks to the Author):

In this interesting work, the authors aimed at estimating the direct and indirect effects of the pandemic on overall mortality in Switzerland. In order to do so, they first estimated the overall mortality in 2020-2022 had the pandemic did not occurred. Then, they used a decomposition model in which the observed overall mortality follows a Poisson distribution with mean being the observed COVID-19 laboratory-confirmed deaths times an estimated parameter called β_1 , and the estimated overall mortality times β_2 . They estimated the direct effects of the COVID-19 pandemic from β_1 and the indirect effects from β_2 .

Although I think the proposed methods are interesting, I have some major concerns about the conclusions and claims and some comments about the models:

- The authors state that, after account for COVID-19 deaths, the observed mortality was lower than expected (lines 23-26, 92-97), but this is not supported by the results considering the credible intervals for β_2 including 1 (0.93;1.01) and for number of deaths compared to the expected (-1776;10700). Therefore, based on the results, there is no evidence of the pandemic having had an indirect effect on mortality. In lines 97-98 the authors briefly acknowledge the wide credible intervals, but elsewhere in the text it is affirmed that the indirect effect was of reduction.
- Could other factors happening during the pandemic period be also impacting the overall mortality? How is that accounted in the model? It would be important to see the percentage of the overall mortality not explained by the pandemic + the expected mortality.
- The correlation between estimated and observed mortality was low for the age groups <70 years, reaching the minimum of around 30% for the age group <40. Good correlation was observed for >70 years old. Since these estimates were used in the decomposition model, the authors could discuss how the different uncertainties by age groups could bias/impact the results. This should also be included in the limitations.
- Lines 125-127 "Detailed data... allowed us to estimate what mortality would have been 2020 to 2022 had the pandemic not occurred" It seems like a strong affirmation considering the poor correlation between estimated and observed mortality for age groups <70 years.
- Lines 129-131: It needs to be better justified considering only temperature, holidays and population as the most important determinants for all-cause mortality.

Reviewer #2 (Remarks to the Author):

General comments:

This report provides a methodologically robust estimate of excess mortality in Switzerland during the COVID-19 pandemic and provides a novel approach for estimating the extent to which excess mortality consists of COVID-19 mortality, misclassified COVID-19 mortality and the indirect effects of the pandemic. It contributes to the wider literature around the impact of the COVID-19 pandemic and contributes significantly to approaches in counterfactual analysis by proposing a novel approach to understanding excess mortality.

The paper is clearly written and contains sufficient supplement material for methodological transparency and reproducibility.

It is not clear how the results are interpreted in relation to mortality displacement. The limitations to interpretation caused by mortality displacement are acknowledged in line 52, then again in line 171. However, the results are interpreted as though no mortality displacement has occurred, with the rationale that the deficit seen in this report is among the young. However, the report does not provide evidence that mortality displacement hasn't occurred, therefore please interpret the findings among the older groups in relation to the unmeasured mortality displacement that has likely occurred among these groups - i.e., expected mortality levels are likely to be overestimated during the latter periods.

In relation to the above - line 80 interprets the results in phase 4 as mortality displacement. Can

you clarify how you can differentiate this deficit as mortality displacement, rather than the protective effect of the pandemic?
it would be useful to have a timeline of mobility restrictions in Switzerland to help with interpretation.

Line 138 you acknowledge the limitation of misclassification of COVID-19 mortality given the report assumes any death following a positive test is a COVID-19 death, then quickly reference autopsies. Surely this is dependent on the length of your study period?
In addition, you do not interpret the potential misclassification in relation to the impact on your conclusions - especially given that this is likely to occur more often among older age groups. Please add this detail.

Specific minor comments:

Consider using an alternative term to 'decompose' when describing the methodology, given the use of the verb in relation to mortality (e.g., decomposing dead bodies!).

Line 52- please qualify how mortality displacement limits the interpretability of results.

I suggest acknowledging reduced circulation of influenza - and the effect of this on non-covid-19 deficits among the elderly.

Please state whether there was any sensitivity analysis of the baseline period of 10 years for the excess deaths model.

line 174- please discuss ascertainment in relation to access to testing during the early pandemic period in Switzerland.

Are the methods intended to be on line 203, or earlier in the paper?

Line 207- is this date of death or date of registration?

Line 210 - a 24-month delay on cause of death detail is considerably larger than many other countries in Europe. Please provide a reference

Line 220- were these phases defined by case rates or death rates?

Reviewer #3 (Remarks to the Author):

Dear Authors

Thank you for the opportunity to read and review your paper: "Direct and indirect effects of the COVID-19 pandemic on mortality in Switzerland".

Overall the paper addresses a very important public health problem: to what extent did the pandemic and the associated control measures impact on total mortality? The methodological approach is innovative and interesting, and the paper is well written, referenced and clearly described.

I do have some comments that I hope might be helpful in redrafting:

Introduction

I1. In the introduction you describe the range of indirect mechanisms that may have had positive or negative impacts on population health. One notable absent mechanism is economic exposures, operating through the restrictions that were placed on economic activity, and the consequent

implications for employment, wages (and associated policy responses such as wage or social security subsidies). Could some text describing this important mechanism be included here? I2. It would be helpful to further clarify in the introduction how direct and indirect COVID-19 deaths are conceptualised for this study. In particular I am thinking of deaths that would have been listed as having a primary cause of death that on face value would be unrelated (e.g. myocardial infarction or stroke), but which may have been triggered by concurrent or recent COVID-10 infection.

Discussion

D1. A limitation of this study is that the full extent of the indirect mortality may not yet have been realised (e.g. through 'long-COVID' mechanisms, unemployment, etc.). Some discussion about the potential for different time lags for direct and indirect mortality, and the need for ongoing surveillance and research on this might be merited.

Methods

M1. If I understand your methods correctly (and I am not a statistician nor have I extensively used Bayesian methods), the key assumption in your model is the estimation of the ratio between laboratory-confirmed COVID-19 deaths and actual direct COVID-19 deaths, derived from a decomposition of the variation in all-cause mortality and the variation in laboratory-confirmed COVID-19 deaths. If so, is there a critical assumption that this ratio is stable over the time period? And following from that, how likely is that assumption to be true? In the UK context, the availability and use of laboratory testing over time has changed markedly, and this would make any such assumption invalid.

I hope these comments are helpful.

Reviewer #1 (Remarks to the Author):

In this interesting work, the authors aimed at estimating the direct and indirect effects of the pandemic on overall mortality in Switzerland. In order to do so, they first estimated the overall mortality in 2020-2022 had the pandemic did not occurred. Then, they used a decomposition model in wich the observed overall mortality follows a Poisson distribution with mean being the observed COVID-19 laboratory-confirmed deaths times an estimated parameter called beta1, and the estimated overall mortality times beta2. They estimated the direct effects of the COVID-19 pandemic from beta1 and the indirect effects from beta2.

> We thank the reviewer for their interest and the helpful comments.

Although I think the proposed methods are interesting, I have some major concerns about the conclusions and claims and some comments about the models:

1A - The authors state that, after account for COVID-19 deaths, the observed mortality was lower than expected (lines 23-26, 92-97), but this is not supported by the results considering the credible intervals for beta2 including 1 (0.93;1.01) and for number of deaths compared to the expected (-1776;10700). Therefore, based on the results, there is no evidence of the pandemic having had an indirect effect on mortality. In lines 97-98 the authors briefly acknowledge the wide credible intervals, but elsewhere in the text it is affirmed that the indirect effect was of reduction.

> We thank the reviewer for this comment. The 95% uncertainty interval around β_2 does indeed include 1. In a statistical hypothesis testing framework, we indeed cannot conclude that the pandemic had a statistically significant indirect protective effect on mortality.

However, the fact that our uncertainty interval includes 1 does not mean that we should not discuss the practical implications of the values of β_2 that are most compatible with the data. This approach is encouraged more and more in the statistical community, as opposed to the dichotomization of estimates into statements of significance¹. In our case, our estimates

favour the interpretation that overall the measures taken to control the pandemic had an indirect protective effect on mortality from other causes, particularly in younger age groups, even though some uncertainty remains.

Your comment made us realise that we had presented the results from the sensitivity analysis, rather than the main analysis. We have rephrased as follows, now presenting the correct results:

Abstract, lines 21-23: *After accounting for COVID-19 deaths, the observed mortality was -4% (95% CrI: -8 to 0) lower than expected. The deficit in mortality was concentrated in age groups 40-59 (-12%, 95%CrI: -19 to -5) and 60-69 (-8%, 95%CrI: -15 to -2).*

Results, lines 113-116: *The relative deficit in all-cause deaths (β_2) was more pronounced in age groups 40-59 and 60-69 with estimates of -12% (95%CrI: -19 to -5) and -8% (95%CrI: -15 to -2), respectively. The deficit was also concentrated in phases 1, 3 and 4 (Figure 3C).*

In the Discussion section, we removed the reference to the absolute deficit of deaths (“estimated at around 4,000 fewer deaths than expected”) and only mentioned the beneficial effect on mortality in age groups 40-59 and 60-69.

Discussion, lines 130-131: *“Finally, the pandemic may have had an indirect beneficial effect on mortality concentrated among age groups 40 to 69.”*

1B - Could other factors happening during the pandemic period be also impacting the overall mortality? How is that accounted in the model? It would be important to see the percentage of the overall mortality not explained by the pandemic + the expected mortality.

> We thank the reviewer for this remark. Indeed, we did not address this sufficiently in the manuscript. The estimates of excess mortality in this paper (as in most others, see for instance²) rely on counterfactual reasoning³, comparing the observed number of deaths with the expected number had the pandemic not occurred. We used information from the 9 years preceding the pandemic (2011-2019) and adjusted for the most critical known predictors of

mortality, including ambient temperature and holidays⁴⁻⁷, assuming that the only difference between 2011-2019 and 2020-2022 is the presence or the absence of the COVID-19 pandemic. Any difference in mortality between the two periods in the adjusted analysis is assumed to be due to the impact of the pandemic. This includes the direct impact of SARS-CoV-2 infections and the effect of control measures taken.

We did not account for changes in mortality that were independent of the pandemic and not observed during the five preceding years. The heatwave during summer 2022 is a case in point: because it would be difficult with this framework to disentangle excesses caused by the unusual heatwave and COVID-19, we decided not to update our analysis to include summer 2022. We revised to clarify this as follows:

Introduction, lines 72-75: *We then developed a method to partition all-cause mortality into deaths directly attributable to SARS-CoV-2 infection and deaths indirectly attributable to the pandemic as a result of all the changes in health, health care, and living or working conditions associated with the SARS-CoV-2 pandemic in Switzerland.*

Discussion, lines 145-152: *As with other studies of excess mortality^{2,8-17}, we assume that the only difference between the 2015-2019 and 2020-2022 time periods other than these covariates is the presence or absence of the COVID-19 pandemic. Therefore, any difference in mortality between these two periods beyond random variation is attributed to the pandemic, either directly or as a consequence of changes in health, health care, and living or working conditions associated with the SARS-CoV-2 pandemic in Switzerland. We ignored the potential impact of other factors on mortality that may have occurred in 2020-2022 but not in 2011-2019. In particular, the study period excluded the heat wave in the summer of 2022.*

1C - The correlation between estimated and observed mortality was low for the age groups <70 years, reaching the minimum of around 30% for the age group <40. Good correlation

was observed for >70 years old. Since these estimates were used in the decomposition model, the authors could discuss how the different uncertainties by age groups could bias/impact the results. This should also be included in the limitations.

> Thank you. We assume that this comment refers to the cross-validation of the expected mortality level (supplementary text, section 1.2.3), and shows the correlation between predicted and observed weekly and the cantonal number of deaths. Please note that correlation is not an ideal metric when counts are small, in our study in the case of people aged <40. Thus, apart from the correlation, we calculated the bias and coverage proportion to gauge the validity of the predictions. We make this point clearer in the Online Supplement:

Supplement, section S1.2.3: *As applying the Pearson correlation coefficient is less appropriate when counts are small; we also calculated the coverage proportion and bias.*

1D - Lines 125-127 "Detailed data... allowed us to estimate what mortality would have been 2020 to 2022 had the pandemic not occurred" It seems like a strong affirmation considering the poor correlation between estimated and observed mortality for age groups <70 years.

> As mentioned above we used 3 different metrics to assess the predictive ability of our results. Pearson correlation is the weakest as it is not suitable for small counts, which probably explains the worse performance in the younger age groups. Nevertheless, having a high coverage and almost 0 bias are strong arguments in favour of our model's predictive ability. We reformulate accordingly:

Methods, lines 295-297: *The weaker correlation in the younger age groups was expected as Pearson correlation is not suitable for small counts. Coverage and bias indicate high predictive accuracy also in these age groups.*

1E - Lines 129-131: It needs to be better justified considering only temperature, holidays and population as the most important determinants for all-cause mortality.

> We thank the reviewer for the comment. Temperature, holidays and population structure and trends might not be the most important determinants of all-cause mortality, but they are certainly among the most important determinants of changes in all-cause mortality over time⁴⁻⁷. Besides, our model also takes into account unknown factors that vary in space and in time (both in seasonal and long-term means), and the uncertainty about these unknown factors is propagated all the way to the final estimates. The predictive ability of the model was very high in the cross-validation, so we did not investigate means to improve it. We justify our approach in the main text as follows:

Discussion, line 136-139: *Temperature, holidays, and population are certainly the most important determinants of changes in all-cause mortality⁴⁻⁷, but the modeling framework we developed also accounts for unknown factors that may vary in space and in time (both in the seasonal and long-term)^{12,18}, resulting in a model with high predictive ability (Supplementary Text S1.1-1.2).*

Reviewer #2 (Remarks to the Author):

General comments:

This report provides a methodologically robust estimate of excess mortality in Switzerland during the COVID-19 pandemic and provides a novel approach for estimating the extent to which excess mortality consists of COVID-19 mortality, misclassified COVID-19 mortality and the indirect effects of the pandemic. It contributes to the wider literature around the impact of the COVID-19 pandemic and contributes significantly to approaches in counterfactual analysis by proposing a novel approach to understanding excess mortality. The paper is clearly written and contains sufficient supplement material for methodological transparency and reproducibility.

> We thank the reviewer for their interest and the constructive comments.

2A - It is not clear how the results are interpreted in relation to mortality displacement. The limitations to interpretation caused by mortality displacement are acknowledged in line 52, then again in line 171. However, the results are interpreted as though no mortality displacement has occurred, with the rationale that the deficit seen in this report is among the young.

However, the report does not provide evidence that mortality displacement hasn't occurred, therefore please interpret the findings among the older groups in relation to the unmeasured mortality displacement that has likely occurred among these groups - i.e., expected mortality levels are likely to be overestimated during the latter periods.

> Thank you. We agree with the reviewer that the way mortality displacement was discussed may have been confusing. We need to differentiate between two phenomena. First, COVID-19 may precipitate deaths that would have anyway occurred in the following days or weeks (this is what is commonly referred to as “harvesting” or “displacement”). This can translate into periods of negative excess following large waves of excess mortality. Second, the

standard approach to compute excess deaths ignores the changes in the population caused by the excess deaths themselves, resulting in an overestimation of the expected number of deaths towards the end of the study period, and thus an underestimation of excess mortality.

Regarding this latter issue, we conducted a sensitivity analysis where the population was corrected by the excess over time. With this correction, we found higher estimates of excess mortality (as the reviewer guessed). However, the estimates of β_1 and β_2 were very similar. We adapted the manuscript as follows (see also new section 1.4 in the supplementary text):

Methods, lines 302-305: *One limitation of the proposed approach is that it ignores changes in the population structure caused by the excess deaths themselves, leading to an overestimation of population and expected deaths, and thus an underestimation of excess towards the end of the study period. We corrected for this bias in a sensitivity analysis (Supplementary text S1.4).*

Results, lines 118-120: *We conducted a sensitivity analysis where the population was corrected by excess mortality. This resulted in higher estimates of excess mortality, with about 1,000 additional excess deaths but had little effect on the estimates of β_1 and β_2 (Supplementary text S1.3).*

Discussion, lines 152-158: *Our modelling framework cannot easily distinguish between reductions in mortality and mortality displacement between epidemic phases. The standard approach to computing excess mortality ignores the changes in population caused by the excess deaths themselves, resulting in an overestimation of the expected number of deaths towards the end of the study period, and thus an underestimation of excess mortality. A sensitivity analysis correcting for this issue resulted in an additional 1,000 excess deaths over the full period, but this did not impact our estimates of β_1 and β_2 .*

We also amended the discussion of mortality displacement in another paragraph:

Discussion, lines 207-211: *The short period of negative excess mortality observed at the beginning of phase 4, just after the largest epidemic wave, could be explained by this harvesting effect. However, this can only be a partial explanation: the overall deficit of deaths was concentrated in the younger age groups and not in the over 70 years old, where mortality displacement typically occurs.*

2B - In relation to the above - line 80 interprets the results in phase 4 as mortality displacement. Can you clarify how you can differentiate this deficit as mortality displacement, rather than the protective effect of the pandemic?

> We agree with the reviewer that this sentence was confusing. In addition, interpretations do not belong to the results section. We replaced the sentence in the results:

Results, line 85: *There were slightly fewer all-cause deaths than expected during phase 4*

And this point is now part of the discussion:

Discussion, lines 207-209: *The short period of negative excess mortality observed at the beginning of phase 4, just after the largest epidemic wave, could be explained by this harvesting effect.*

2C - it would be useful to have a timeline of mobility restrictions in Switzerland to help with interpretation.

> We thank the reviewer for the suggestion and added a timeline of non-pharmaceutical interventions in figure 1C. We refer to this figure in the discussion.

Line 138 you acknowledge the limitation of misclassification of COVID-19 mortality given the report assumes any death following a positive test is a COVID-19 death, then quickly reference autopsies. Surely this is dependent on the length of your study period? In addition, you do not interpret the potential misclassification in relation to the impact on your conclusions - especially given that this is likely to occur more often among older age groups. Please add this detail.

> We agree with the reviewer and clarified as follows:

Discussion, lines 163-170: *We assumed deaths with a positive SARS-CoV-2 test were caused by COVID-19, although the infection could be incidental in some cases (the median delay from test to deaths was 10 days, interquartile range 6 to 15). Such misclassification would only concern a small proportion of deaths with laboratory-confirmed infection, as was shown in cause-of-death data¹⁹. Autopsies of patients dying in hospital following a positive SARS-CoV-2 test also suggest that during the study period, causes of death were generally directly related to COVID-19²⁰. Still, this potential bias may lead to an underestimation of the ascertainment proportion in our analysis.*

Specific minor comments:

2D - Consider using an alternative term to 'decompose' when describing the methodology, given the use of the verb in relation to mortality (e.g., decomposing dead bodies!).

> We thank the reviewer for pointing out this unfortunate double meaning and replaced “decompose” by “partition” or “separation” throughout the manuscript and online supplement.

2E - Line 52- please qualify how mortality displacement limits the interpretability of results.

> We now address in detail how mortality displacement affects our results in the Discussion section and consider that adding this point in the introduction would not be appropriate.

2F - I suggest acknowledging reduced circulation of influenza - and the effect of this on non-covid-19 deficits among the elderly.

> We agree. Influenza was mentioned in the discussion, but we added the following point to further clarify:

Discussion, line 214-219: *The concentration of the mortality deficit in the younger age groups also argues against an important role of a reduced prevalence of other pathogens. For example, influenza leads to mortality in the older age groups, the lack thereof would therefore be expected to result in a mortality deficit in these age groups. Influenza seasons during 2011-2019 were accounted for by the spatio-temporal component of the model.*

2G - Please state whether there was any sensitivity analysis of the baseline period of 10 years for the excess deaths model.

> We did not conduct sensitivity analyses on the formulation of the excess death model, but evaluated its performance on the baseline period using cross-validation. We reformulate the text to highlight this:

Methods, lines 293-297: *The model has been internally validated and found to be unbiased and to have high predictive accuracy. The weaker correlation in the younger age groups was*

expected as Pearson correlation is not suitable for small counts. Coverage and bias indicate high predictive accuracy also in these age groups.

2H - line 174- please discuss ascertainment in relation to access to testing during the early pandemic period in Switzerland.

> We changed the text as follows:

Discussion, lines 198-204: *During the first epidemic phase, testing capacities were limited which might have affected ascertainment of COVID-19-related deaths. However, β_1 in epidemic phase 1 did not substantially differ from the overall estimate. While testing increased over time in Switzerland, the targeting of tests to vulnerable population groups may have simultaneously decreased, so that the ascertainment of severe cases remained relatively stable in the first 4 phases. The lower ascertainment towards the end of the study period might be explained by reduced testing once vaccines became available.*

2I - Are the methods intended to be on line 203, or earlier in the paper?

> The methods part was put at the end in accordance with Nature Communications guidelines.

2J - Line 207- is this date of death or date of registration?

> Thank you for spotting this potential confusion, deaths were aggregated based on the date of death. We made the following clarification in the text:

Methods, lines 251-252: *... counts of deaths from any cause by age, sex, and canton for each week using the date of death from 2011 to 2019.*

2K - Line 210 - a 24-month delay on cause of death detail is considerably larger than many other countries in Europe. Please provide a reference

> Data on cause of deaths for 2020 were published in September 2022, during the review process. These data are in accordance with our findings, and were added to the discussion:

Discussion, lines 191-195: “Recently-published data on causes of deaths in 2020 and the first half of 2021 provided a partial external validation of our results¹⁹. According to these data, the number of deaths with COVID-19 as a cause was 2,142 from February to May 2020 (24% higher than laboratory-confirmed deaths) and 10,650 from September 2020 to February 2021 (36% higher than laboratory-confirmed deaths).”

2L - Line 220- were these phases defined by case rates or death rates?

> The time periods were defined by the FOPH using the lowest points in reported cases. We added this detail in the text:

Methods, line 262: Dates were grouped into seven epidemic phases by the FOPH *using the lowest counts of reported cases...*

Reviewer #3 (Remarks to the Author):

Dear Authors

Thank you for the opportunity to read and review your paper: "Direct and indirect effects of the COVID-19 pandemic on mortality in Switzerland".

Overall the paper addresses a very important public health problem: to what extent did the pandemic and the associated control measures impact on total mortality? The methodological approach is innovative and interesting, and the paper is well written, referenced and clearly described.

I do have some comments that I hope might be helpful in redrafting:

> We thank the reviewer for the encouraging and constructive comments.

Introduction

3A - In the introduction you describe the range of indirect mechanisms that may have had positive or negative impacts on population health. One notable absent mechanism is economic exposures, operating through the restrictions that were placed on economic activity, and the consequent implications for employment, wages (and associated policy responses such as wage or social security subsidies). Could some text describing this important mechanism be included here?

> We thank the reviewer for pointing this out. Please note that Switzerland put in place a generous short-time work support scheme to prevent redundancies, which benefited many workers (1.3 million at the peak in April 2020). The government spent more than 10 billion Swiss francs on the scheme. As a consequence, the increase in unemployment in 2020 was relatively modest (from 2.3% in 2019 to 3.1%) and the unemployment rate decreased again in the following years. Nevertheless, unemployment has negative consequences for the individuals concerned as well as for the economy and society. We now acknowledge this in the introduction as follows:

Introduction, line 72-75: *We then developed a method to partition all-cause mortality into deaths directly attributable to SARS-CoV-2 infection and deaths indirectly attributable to the pandemic as a result of all the changes in health, health care, and living or working conditions associated with the SARS-CoV-2 pandemic in Switzerland*

Discussion, lines 220-227: *However, we cannot rule out harmful effects, such as delay or avoidance of medical care^{21,22}, increases in drug use and suicidal ideation^{23,24}, increases in interpersonal violence²⁵ and other risks associated with economically precarious living conditions. These effects were likely mitigated by government policies. The short-time work support scheme, which benefited many workers (1.3 million at the peak in April 2020), probably prevented many redundancies. Indeed, the increase in unemployment in 2020 was relatively modest (from 2.3% in 2019 to 3.1%), and the unemployment rate decreased again in the following years^{26,27}.*

3B - It would be helpful to further clarify in the introduction how direct and indirect COVID-19 deaths are conceptualised for this study. In particular I am thinking of deaths that would have been listed as having a primary cause of death that on face value would be unrelated (e.g. myocardial infarction or stroke), but which may have been triggered by concurrent or recent COVID-10 infection.

> We followed the advice of the reviewer and added a clarification in the introduction:

Introduction, lines 34-39: *Deaths directly attributable to SARS-CoV-2 infection may be both laboratory-confirmed deaths and deaths attributable to SARS-CoV-2 without testing (e.g., deaths following a heart attack or stroke^{28,29}). Combined with the high transmissibility of SARS-CoV-2, this has resulted in more than 6.5 million laboratory-confirmed deaths worldwide (as of October 10, 2022)³⁰, which is likely an underestimate of the true number of deaths directly attributable to SARS-CoV-2 infection.*

Discussion

3C - A limitation of this study is that the full extent of the indirect mortality may not yet have been realised (e.g. through 'long-COVID' mechanisms, unemployment, etc.). Some discussion about the potential for different time lags for direct and indirect mortality, and the need for ongoing surveillance and research on this might be merited.

> We thank the reviewer for this comment. We made the following changes in the text:

Discussion, lines 161-163: *We also only considered the short-term effects of COVID-19 on mortality, while SARS-CoV-2 infection may lead to increased mortality from cardiovascular, cancer or respiratory system cause of death in the following months²⁹.*

Methods

3D - If I understand your methods correctly (and I am not a statistician nor have I extensively used Bayesian methods), the key assumption in your model is the estimation of the ratio between laboratory-confirmed COVID-19 deaths and actual direct COVID-19 deaths, derived from a decomposition of the variation in all-cause mortality and the variation in laboratory-confirmed COVID-19 deaths. If so, is there a critical assumption that this ratio is stable over the time period? And following from that, how likely is that assumption to be true? In the UK context, the availability and use of laboratory testing over time has changed markedly, and this would make any such assumption invalid.

> The reviewer correctly understood that our model estimates the ratio between laboratory-confirmed deaths and direct COVID-19 deaths. However, our approach does not require any stability of this ratio over time, in general, or across the analysis periods, but rather estimates the weighted average of the ratio over the period considered. The availability and targeting of tests has indeed changed in Switzerland, which is the reason why we looked at estimates of β_1 per time period. We added this point to the discussion:

Discussion, line 198-204: *During the first epidemic phase, testing capacities were limited which might have affected ascertainment of COVID-19-related deaths. However, β_1 in epidemic phase 1 did not substantially differ from the overall estimate. While testing increased over time in Switzerland, the targeting of tests to vulnerable population groups may have simultaneously decreased, so that the ascertainment of severe cases remained relatively stable in the first 4 phases. The lower ascertainment towards the end of the study period might be explained by reduced testing once vaccines became available.*

I hope these comments are helpful.

> Yes, many thanks.

References

- 1 Amrhein, V., Greenland, S. & McShane, B. (Nature Publishing Group, 2019).
- 2 Islam, N. et al. Excess deaths associated with covid-19 pandemic in 2020: age and sex disaggregated time series analysis in 29 high income countries. *bmj* 373 (2021).
- 3 Hernán, M. A. & Robins, J. M. (CRC Boca Raton, FL, 2010).
- 4 Haines, A., Kovats, R. S., Campbell-Lendrum, D. & Corvalán, C. Climate change and human health: impacts, vulnerability and public health. *Public health* 120, 585-596 (2006).
- 5 Phillips, D. P., Jarvinen, J. R., Abramson, I. S. & Phillips, R. R. Cardiac mortality is higher around Christmas and New Year's than at any other time: the holidays as a risk factor for death. *Circulation* 110, 3781-3788 (2004).
- 6 Thommen, O. Heat wave 2003 and mortality in Switzerland. *Swiss medical weekly* 135 (2005).
- 7 Walker, A. S. et al. Mortality risks associated with emergency admissions during weekends and public holidays: an analysis of electronic health records. *The Lancet* 390, 62-72 (2017).
- 8 Cronin, C. J. & Evans, W. N. Excess mortality from COVID and non-COVID causes in minority populations. *Proceedings of the National Academy of Sciences* 118, e2101386118 (2021).
- 9 Economist, T. & Solstad, S. The pandemic's true death toll. *The Economist* (2021).
- 10 Heuveline, P. The COVID-19 pandemic adds another 200,000 deaths (50%) to the annual toll of excess mortality in the United States. *Proceedings of the National Academy of Sciences* 118, e2107590118 (2021).
- 11 Karlinsky, A. & Kobak, D. Tracking excess mortality across countries during the COVID-19 pandemic with the World Mortality Dataset. *Elife* 10 (2021).

- 12 Konstantinoudis, G. et al. Regional excess mortality during the 2020 COVID-19 pandemic in five European countries. *Nature communications* 13, 1-11 (2022).
- 13 Organization, W. H. (January, 2022).
- 14 Staub, K. et al. Historically high excess mortality during the COVID-19 pandemic in Switzerland, Sweden, and Spain. *Annals of internal medicine* 175, 523-532 (2022).
- 15 Wang, H. et al. Estimating excess mortality due to the COVID-19 pandemic: a systematic analysis of COVID-19-related mortality, 2020–21. *The Lancet* 399, 1513-1536 (2022).
- 16 Weitkunat, R., Junker, C., Caviezel, S. & Fehst, K. Mortality monitoring in Switzerland. *Swiss Medical Weekly* (2021).
- 17 Whittaker, C. et al. Under-reporting of deaths limits our understanding of true burden of covid-19. *Bmj* 375 (2021).
- 18 Kontis, V. et al. Magnitude, demographics and dynamics of the effect of the first wave of the COVID-19 pandemic on all-cause mortality in 21 industrialized countries. *Nature medicine* 26, 1919-1928 (2020).
- 19 Office, F. S. Specific causes of deaths.
- 20 Elezkurtaj, S. et al. Causes of death and comorbidities in hospitalized patients with COVID-19. *Scientific Reports* 11, 1-9 (2021).
- 21 Czeisler, M. É. et al. Delay or avoidance of medical care because of COVID-19–related concerns—United States, June 2020. *Morbidity and mortality weekly report* 69, 1250 (2020).
- 22 Riera, R. et al. Delays and disruptions in cancer health care due to COVID-19 pandemic: systematic review. *JCO Global Oncology* 7, 311-323 (2021).

- 23 Czeisler, M. É. et al. Mental health, substance use, and suicidal ideation during a prolonged COVID-19-related lockdown in a region with low SARS-CoV-2 prevalence. *Journal of psychiatric research* 140, 533-544 (2021).
- 24 Zaami, S., Marinelli, E. & Vari, M. R. New trends of substance abuse during COVID-19 pandemic: an international perspective. *Frontiers in Psychiatry* 11, 700 (2020).
- 25 Mazza, M., Marano, G., Lai, C., Janiri, L. & Sani, G. Danger in danger: Interpersonal violence during COVID-19 quarantine. *Psychiatry research* 289, 113046 (2020).
- 26 Müller, T., Schulten, T. & Drahokoupil, J. Job retention schemes in Europe during the COVID-19 pandemic – different shapes and sizes and the role of collective bargaining. *Transfer: European Review of Labour and Research* 28, 247-265, doi:10.1177/10242589221089808 (2022).
- 27 Office, F. S. Labour market indicators 2022 - Comments on findings.
- 28 Raisi-Estabragh, Z. et al. Cardiovascular disease and mortality sequelae of COVID-19 in the UK Biobank. *Heart* (2022).
- 29 Uusküla, A. et al. Long-term mortality following SARS-CoV-2 infection: A national cohort study from Estonia. *The Lancet Regional Health-Europe*, 100394 (2022).
- 30 Worldometer, D. COVID-19 coronavirus pandemic. World Health Organization, www.worldometers.info/coronavirus (2020).

REVIEWERS' COMMENTS

Reviewer #2 (Remarks to the Author):

Thank you, the responses adequately address my comments.

Reviewer #3 (Remarks to the Author):

Thank you for the responses and revisions to your manuscript following the initial peer review comments. I think you have very expertly dealt with all of these.